# Father's Use of Parental Leave in Organizations with Different Institutional Logics

**Lisa Vaagan Moen, Elin Kvande * and Kine Nordli**

Department of Sociology and Political Science, Norwegian University of Science and Technology, NO-7491 Trondheim, Norway; lisa.v.moen@ntnu.no (L.V.-M.); knordli91@gmail.com (K.N.)

* Correspondence: elin.kvande@ntnu.no

**Abstract:** Although the use of the father's quota of parental leave has become a majority practice among Norwegian fathers, there is some variation between different groups of fathers. This article explores how male managers in the engineering industry and male brokers in the finance industry use the father's quota. Based on the theoretical framework of institutional logics, the article uses two pairs of opposite concepts-'available and unavailable' and 'replaceable and irreplaceable' in a work context, to focus on how the use of the father's quota is affected. Analyzing two different male-dominated organizations, the findings show how the use of the father's quota depends on different institutional logics, which sets the framework for the practice and culture of the two organizations. The male managers in the engineering industry become unavailable and replaceable in their organizations, thus making it possible for the fathers to use the father's quota and parental leave. In contrast to this, the institutional logic in the finance industry makes brokers available and irreplaceable in their organizations, thus making it difficult for them to use father's quota or parental leave

**Keywords:** fathers; parental leave; careers; institutional logics

## 1. Introduction

In the Nordic welfare states, where family policies are no longer based on the male provider model but are rather based on both parents being providers and caregivers, there may be a greater acceptance and use of parental leave than in other countries (Brandth and Kvande 2019a). Norway introduced a father's quota of four weeks into the parental leave scheme in 1993. Research has shown that the father's quota, which is currently 15 weeks, has worked well as a measure to get fathers to take parental leave. Fathers' use of the father's quota is supported and respected by their employers (Brandth and Kvande 2019b). One important reason for this is that the father's quota is an individual and earmarked right for the father.

Although the use of the father's quota has become a majority practice among Norwegian fathers, there is some variation between different groups of fathers. Highly educated fathers have a more positive orientation toward the father's quota than other fathers (Lappegård and Bringedal 2013), and the actual use of leave by this group of fathers is also highest. Fathers with very high incomes are, however, among those who use the father's quota the least (Grambo and Myklebø 2009; Kitterød et al. 2017). Career fathers with elite education view their jobs as an investment in career development; for them, taking advantage of the father's quota may have indirect and long-term consequences that can reduce career opportunities (Halrynjo and Lyng 2013). This may in turn explain the lower use of leave in this group.

The use of the father's quota may also vary from one work organization to the next. Nordberg (2019) shows that one must see parental leave in connection with organizations' different logics, goals and strategies. The purpose of her study was to investigate how managers in the police and legal profession

facilitated employee's use of care rights. She finds that different logics set the framework for managers' professional expectations and stipulate how they relate to employees' use of care rights.

The point of departure for this article is the significant changes that have taken place in Norwegian society when it comes to the participation of fathers in providing care for their children. Based on an interview study of male managers in the engineering industry and male brokers in the finance industry, this article explores what experiences these fathers have had when it comes to the use of parental leave and the father's quota. The article will therefore to answer the following research questions: How do the different institutional logics in organizations affect father's use of parental leave?

## 2. Implementation of Parental Leave in Working Life

Previous study has examined whether the use of leave by fathers in career professions may have indirect and long-term detrimental effects on their careers (Halrynjo and Lyng 2013). Data derived from a questionnaire study of lawyers, business economists, and graduate engineers from 2007, show that the father's quota has had a great impact on fathers' use of available leave time in the elite professions in general. After the father's quota was introduced in 1993, 71 percent of fathers took parental leave compared to 15 percent before the father's quota was introduced. When it comes to the link between career development and use of leave in this group, the Halrynjo and Lyng find that the use of parental leave does not influence the probability of fathers being promoted to middle-management positions. Nevertheless, they find that the probability of reaching senior managerial positions decreases with the use of such leave.

They explain this by stating that parents in career professions are facing a dilemma when using parental leave, which is based on what they call a facilitation logic for parents, meaning that the right to leave time can clash with a career logic that demands continuous input and performance (Halrynjo and Lyng 2013). Career jobs are often characterized by individual competition where visibility and performance can provide an important competitive edge. Therefore, family policy programs introduced to improve the possibility of combining work and family for fathers may have a counterproductive effect on this group of fathers, according to Halrynjo and Lyng (2013). In the competition with other colleagues they may risk being demoted from the A team, dropping down to the B team because they have less time available for work if they go on the leave.

The challenges are thus connected to the risk of being 'replaceable' in a career path demanding that one is 'irreplaceable' by being available, according to Halrynjo and Lyng (2013). Even the earmarked father's quota may be negotiable in such a context. This is because the effect of the earmarking is undermined in career jobs where choosing not to use one's legal rights becomes a symbolic marker of dedication to job and career (Lyng 2010). Although parental leave is a one-time and time-limited absence, when many highly qualified and motivated colleagues are competing for the same opportunities, projects, and promotions, even a short-term absence, such as the father's quota, may leave a male employee lagging behind. The studies were, however, conducted from 2005 to 2007. Over the last decade, the use of the father's quota has become an accepted norm among the majority of Norwegian fathers (Kitterød et al. 2017). This may have impacted both the understanding and use of the father's quota.

In a new report, Halrynjo et al. (2019) have explored gender differences in the financial industry based on interviews with both women and men. The purpose was to contribute new knowledge about culture, career choices, opportunities, and limitations, the importance of children and care responsibility and valuation of competence and responsibility. The report shows that men and women in finance make different career choices governed by different conditions. Parental leave and absence from work are especially pinpointed as risks in Front End Finance. This is because one has a lot to lose in terms of customers, salary development, bonuses, and career opportunities. Fathers can postpone, divide, and adapt parental leave to the needs of their clients, thus retaining their clients and portfolio(s) (Halrynjo et al. 2019). Mothers must take leave when the child comes and give up clients

and portfolio(s) to colleagues who can follow up on customers. Thus, mothers are at a greater risk of losing their customers and "moving back to start."

International research finds that, executive men who are leaders in lucrative industries and have finance related responsibilities encounter the 'schema of work devotion' that promised several intertwining rewards including career advancement, status, financial gain and interesting work (Blair-Loy and Williams 2017). They showed that the emphasis on devotion to work may inhibit executive men from taking time off to care for family, even if they are offered paid parental leave. This is in line with previous research conducted by Blair-Loy and Wharton (2002), where they found that the use of flexible and family-friendly initiatives which entail either a direct or an indirect break with 'the rules of the game' can come at a cost for the employees using them. Part-time, job sharing, and flexibility programs were available in a large American firm, but the awareness of the actual rules of the game meant that these programs were hardly ever used (Hochschild 1997). This was the case even though workers experienced stress and conflict when trying to balance family life and working life. A study of what prevents men from taking parental leave in Canada (McKay and Doucet 2010) points to four factors: the preferences of mothers, no earmarking of the leave, too short leave, and the social norms at the workplace.

Fathers' use of parental leave is generally context-dependent, as it varies according to the country in question and its welfare system, which may have highly unique family policies and working life regulations. This use may also vary from one work organization to the next. Nordberg (2019) explores how managers in the police and legal profession facilitate employees' use of care rights. The aim was to investigate how managers view gender equality when facilitating employees' use of care rights. Semi-structured interviews have been conducted with managers in the police and legal profession on how they respond to employees exercising the right to parental leave and the right to work reduced hours.

The findings show that parental leave must be seen in connection with different logics for which goals and strategies exist in the organization. The manager's understandings are based on different institutional logics that are dominant in the organizations. Institutional logics are ideal type categories for how individuals and organizations' behaviors are influenced by social institutions, in the form of both cultural symbols and material practices that are peculiar to a given institutional order (Thornton et al. 2012). Institutional logics have an impact on how phenomena make sense because they act as frames of reference for what appear to be relevant goals, values, issues, solutions, and practices. In Nordberg's study (2019), these logics set the framework for managers' professional expectations and how they relate to employees' use of care rights.

To analyze how fathers in two different organizations use the father's quota and parental leave, two theoretical concept pairs are central: (1) *replaceable and irreplaceable*, and (2) *available and unavailable*. As shown above, these concepts have been used in analyses of whether the use of leave by fathers in career professions may have indirect and long-term detrimental effects on their careers (Halrynjo and Lyng 2013). To explore how a father's use of the father's quota and parental leave may depend on the type of organization, we apply the term *institutional logic*.

## 3. Method and Design

Based on two datasets on male managers in the engineering industry and on male brokers in the finance industry, we analyzed the informant's use of the father's quota and parental leave. These are both male-dominated fields of work. The interviews were conducted in the autumn of 2016, in the winter of 2017 (Moen 2018) and in the autumn of 2018 (Nordli 2019).

### 3.1. Managers

This study of managers' experiences with the father's quota has used semi-structured interviews with nine male managers within larger engineering companies. These nine fathers were found through strategic selection according to three criteria. Firstly, they were fathers who had management positions.

Secondly, they had used their parental leave. Thirdly, they had children young enough for the fathers to be able to remember the toddler period and how the leave was experienced in relative detail. Another reason for choosing this criterion was that fathers with children between zero and six years of age belonged to the group of fathers who had gone through great changes in terms of the father's role (Kitterød 2013). In some cases, the fathers had older children, and some also had younger children.

The fathers were recruited by applying the snowball method, and all the fathers were very helpful in contacting additional potential interviewees. Pseudonyms are used herein to keep the identity of both fathers and companies anonymous. The interviews, lasting approximately one hour, were held in meeting rooms at their workplaces, except for four interviews that were conducted in their homes or by telephone. The interviews focused on the father's day-to-day life at home and at work. The main topics of the interviews concerned work duties, well-being at work, the leave itself, the combination of work and home, caregiving responsibilities at home and detailed descriptions of 'an ordinary day' at home and at work.

The male managers for the most part belonged to the same professional group, as they had been educated as engineers and graduate engineers. All the informants were middle-managers in their companies, either heads of section or department heads. Their duties in these positions could include human resource management, project management, and team leadership.

### 3.2. Brokers

Semi-structured interviews were conducted with seven informants in the financial industry. These informants were found through a strategic selection through two criteria (Nordli 2019). Firstly, the desire was primarily to talk to fathers who had children aged one to seven years. In this way, they could remember the most from the toddler period. Secondly, the starting point was that the fathers should have used leave, but this was not emphasized in the recruitment of informants.

The informants were recruited through the director of one of the larger brokerage houses that were contacted. The director turned out to be engaged in this topic, which quickly started the process. The director encouraged everyone in the organization who met the requirements to participate in the study. Again, pseudonyms were used for our informants. The informants wanted short interviews; accordingly, they lasted around fifty minutes. The main topics of the interviews concerned career, parental leave, family and the combination of work and home.

The informants were further divided into two groups based on their tasks (Nordli 2019). The first group has been named 'front office' and the second group has been named 'middle office'. This is based on how the informants have outlined the structure of the brokerage. In addition to these two groups, a third grouping, 'back office', was described, but none of the informants fit into this category. In the 'front office' you find brokers. These are informants who have work tasks closely linked to the execution of purchases and sales. The informants in the middle office are often referred to as a supervisory body and will in this study be called consultants. To a large extent, their day is governed by analyses and transactions. In this article, the focus will mainly be on the fathers who belong to the front office group and work as brokers in order best to compare with managers. This is because brokers in the front office group seems to be the group that use the parental leave least.

When analyzing the use of the father's quota among male managers in the engineering industry and male brokers in the finance industry we applied a narrative analysis using the concept pairs 'available and unavailable' and 'replaceable and irreplaceable'. We also use the term 'institutional logics' to see whether the father's use of the father's quota depends on different logics within the two different organizations. The results will be presented separately in the following.

## 4. Managers: Becoming Unavailable and Replaceable at Work

*4.1. "Business Goes on Anyway"*

The interviews show how several of the fathers in the managers group considered themselves irreplaceable in their jobs before taking leave, but this view often changed during their leave. The fathers in this study have a lot of responsibility for their own employees, and the work is mainly characterized by project work. Markus, an engineer and project manager who had used the father's quota had the following to say:

> You might think that you're 'one of a kind', that you're irreplaceable or things won't function at work. And, you might have learnt that if you don't do anything at all chaos will ensue, so that there will be loads of backlog at work when you return. Ten weeks is a long time.

However, he found that both he and the company coped well during his leave. With a surprised tone, he said: "everything didn't come to a standstill, and that's good." Thus, it turned out that he was not quite 'one of a kind' at his workplace after all, but that the company was able to find solutions without him, and colleagues were able to fill his role. He experienced that he was replaceable, and even that the company enjoyed success during the weeks that he was away. Because Markus' workplace found solutions without him, this allowed Markus to take his leave without feeling guilty. In this case, the workplace was never an obstacle for Markus.

Karl, an engineer, had the same experience, when he used 12 weeks of the father's quota, and he explicitly reflected on having felt irreplaceable in the job context:

> So I think it's quite good, both for the fathers and for the families, and maybe even for working life, really, that you find out that you're not irreplaceable, I can be away and the business simply goes on anyway. When you only keep working and working, you make yourself indispensable, you deal with everything yourself. It in a way forces you to stay away for some time.

By making himself unavailable for the job he discovered that the company managed without him; Karl experienced that he was actually not at all irreplaceable at work. He pointed this out when he said: "Business goes on anyway". This quotation also points out that it is possible to influence whether one wants to be replaceable or irreplaceable at work. The more you work, the more irreplaceable you become. Making himself inaccessible during a long continuous father's quota helped Karl to see this in context. He saw it as his own choice to make himself unavailable, and did not see it as a "risk" of being made replaceable.

*4.2. Making Oneself Irreplaceable*

Aron, an engineer, was one of the informants who had used both a continuous and part-time father's quota. With his last child, he used the father's quota of ten consecutive weeks, and he took two extra weeks. His work tasks were many and varied, and could range from project work to management to employees. He stated that his choice was based on the fact that he wanted to make himself less available for the job and therefore he did not choose a flexible part-time quota:

> . . . So I probably felt that if I was to do it (use the part-time father's quota) I would not get much time off. Say that if I had leave two days a week, for example, then I think it would have been difficult for me to turn off completely [ . . . ], so I think that on those two days there would have been a lot of focus on the job, as I would have been available.

His reason for using continuous leave was his concern that he otherwise would not have been off the job enough. Making himself less accessible for the job was important to him if he was to succeed at being fully present in caregiving. This choice was based also on his experiences from his first father's quota leave period when he felt overloaded when he took out the part-time father's quota. "I learned

the hard way the first time [ ... ] when you made yourself accessible, both for colleagues and clients, the phone would ring and emails would come, and then there was too much focus on it. So the first leave was not very good [ ... ]". To manage to set a boundary between the job and caregiving during his second leave he chose to make himself less available for his job and more available for his child.

Similar to Aron, Harald found that he was able to put aside his job during his leave. Harald is a project manager for scientists and technicians in a team, and he told us: "I received the odd phone call which I answered, but there was nothing that was really a bother. I tried to be very aware of it, and I think I succeeded". His focus was on being a caregiver—even with a new managerial position waiting for him. Several of the fathers had a few phone calls from the workplace during their leave, but they spoke about them as being 'pleasant', or not important. Leo chose to make himself unavailable for his job. For him the caregiving quickly turned into regular routines to which he got accustomed, and he was thus quite detached from his job.

Unlike the other interviewed fathers, August used the shared leave in addition to the father's quota, so his leave amounted to a full six months. He had two young children, and during the interview, he was on leave with his youngest child. One of the reasons why he succeeded at being unavailable for his job during his leave was that the project in which he was involved was reaching completion. Thus, it was not necessary to introduce a completely new person in what he called an "intensive and important role"; instead a colleague already working on the project took care of it. Therefore, because August had collaborated with others in the project it was unproblematic for a colleague to take over his duties while he went on leave. In other words, the work tasks made it easier for him to take the father's quota.

August received only the odd phone call with questions to which he needed to respond during his leave. It is also interesting that he said that he had adopted a 'normal working life' situation, which suggests that it was common in his company to take leave. In his company, it was up to each employee to choose whether he or she wanted to be among the 'ambitious ones' or among those with a 'normal working life'. The latter was defined as working 'nine-to-five'. At the same time, this can be interpreted to mean that a 'normal working life' also includes taking leave. Thus, he opted out of the career logic which would have required continuous effort and achievement and involves instead working hours which allow a combination of work and family. In this case, it seems that the organization supports and facilitates both forms of work and it is up to each individual to decide how to proceed. This is the dominant institutional logic in this organization.

### 4.3. The Leave as a Boundary-Setter in Relation to Work

Emil worked with several different projects and he also had a lot of responsibility for his employees in his engineer company. He used two different leaves, one was full-time and unbroken and the other was part-time. The first time he was to use the father's quota, Emil told us that it was acceptable for the company that he used the leave, but that there was no encouragement to use it. Thus, it was accepted, but not more than that. He also encountered comments such as "Doesn't the child have a mother?" and he talked about his workplace as having an 'old-boys' culture. For this reason, Emil did not embark on his leave with the most favorable of conditions from the company's point of view, and like Aron he chose to make himself unavailable during the weeks he used his father's quota full time. "During the first [leave] I didn't work at all, I removed the mail app from my phone too, and then I didn't work at all [ ... ] I didn't want them to rob me of the joy of leave time." He very deliberately made himself unavailable for his job. This was due to negative situations and incidents that had occurred prior to the start of his leave. In this way, Emil's period of leave turned into a protest against the workplace and the prevailing 'old-boys' culture. He adopted what may be called an oppositional role also by having the right and opportunity to take leave, but also by using this opportunity as a protest. In this case, the informant chose to defy the social norms and violate the "prevailing rules of the game" which Blair-Loy and Wharton (2002) describe. Emil goes against the institutional logic within the company in order to take his father's quota.

Markus told that he was able to choose when he wanted to make himself available or inaccessible during his leave time. The period of leave became a boundary-setter for him in the sense that he chose to not accept work duties there and then: " … it felt good to be able to simply say 'no, I'm on leave, so you better come back in a couple of months'." He deliberately used his leave to set a boundary so that he could put work assignments aside. Markus also told us that the period of leave enabled him to "turn down the pressure a notch or two" and be present at home and care for his child.

*4.4. Career Development*

Several of the fathers found that they were neither 'irreplaceable' nor 'indispensable' in their jobs while home on leave. Thus, when in competition with colleagues, using parental leave may mean being demoted from the A team to the B team in a career context. The interview material shows, however, that our informants did not experience this. The perception of being replaceable at work may be connected to how the fathers experienced that the period of leave did not have a detrimental effect on their careers. Halrynjo and Lyng (2013) describe how career fathers may risk being dropped from the A team to the B team due to their absence from work during leave. Our data material shows, however, that the fathers compared returning to work to returning from a long vacation. It must be pointed out that the leave itself is not compared with a vacation; the comparison refers only to the feeling when returning to work. One of the implications of this is that there is accumulated work waiting for them when they return as is the case when returning from an ordinary vacation.

When asked whether the leave had an impact on their career development Leo responded: "No, not at all". He returned to the same work assignments and nothing had changed. This was clearly expressed also by Emil, who found that the leave had not had an impact on his career development, and this was particularly clear in Harald's case. A new position was waiting for Harald when he came back to work after his leave. Therefore, he did not in any way feel that his leave had constrained his career opportunities, but rather he could claim that: " … I was promoted while I was at home." He was given a managerial position with greater responsibility and a higher salary.

Only one of the fathers in this study reported that the leave caused his career to stall. This informant took a total of six months of leave. His feeling that his career was stagnating may derive from the fact that he was absent from work for a relatively long period of time compared to the other fathers in the study. It is important to point out that this father did not experience the stagnation as problematic because he had chosen 'normal working hours' and his career ambitions belonged to the past. This may also tell us something in general about fathers who use leave, as they may possibly have reached a phase in life where their priorities have changed. Their career is in part put on the back burner while being a caregiver rises to top priority.

It is interesting to take a closer look at how managers with the experience of having been home on leave presented their points of view on the father's quota and fathers' leave to their employees. Emil kept his own experience in mind when assessing applications for leave from his employees. He stated: "There's one worker now for whom I'm the supervisor and he's planning to take 26 weeks, that's half of the total leave time to which he is entitled. Of course, he should be able to do this, that's the culture I want to have, that much tolerance." Emil's view on what type of culture he wanted to have in terms of leave for fathers also showed how he as a manager helped to determine the norms in his workplace. He used his experiences from his first leave when he had received negative feedback from the company and did not want his employees to experience the same. Creating this type of culture was something Aron also wanted in the company where he worked: "There is acceptance for taking a large portion of the available leave with no questions asked. We have people who take the minimum and also people who have been away for seven or eight months. I think this is good, and I hope it's a mark of quality for our workplace."

Thus, it was up to the fathers themselves to decide how much of the leave they wanted to take. Having this positive attitude regarding the father's quota and leave was something Aron called a *mark of quality* for the company. This illustrates that taking the father's quota has become a norm in this

company. Edvin, who was the head of a section with responsibility for many employees, shared this attitude. He told us that no questions were raised about his taking the father's quota and leave, and added that: "It's the most natural thing in the world that people use the leave." He and the rest of the company were also interested in making this the basic policy: " . . . the bottom line in the company is that this program is positive, and all the managers take leave". Therefore, it was not a problem to go on leave, even for the heads of the company.

## 5. Brokers: Becoming Available and Irreplaceable at Work

### 5.1. "No Matter What People Actually Say"

The brokers experience a great deal of pressure from the job, and networking and contact with customers is important in their everyday work. In other words, the brokers need to have a good network to succeed. Based on this, it is interesting to see how the brokers choose to use parental leave and the reasons behind the different choices.

Per was a new employee and broker in the current company, and therefore he used the father's quota in his former workplace. At this workplace, he took out ten weeks, which was the quota at that time. He experienced a supportive culture for taking leave at his former workplace. Per points out that it is important to take leave in regards to his contact with the child, and he experienced his leave as a cozy and relaxing time with his child.

> I am going to take it this time too (the father quota), I feel like there is something you need to do to get a connection with your kid. Therefore, I am going to do that here too, no matter what people actually say. I do not care . . . if it is not ok, then it is too bad for them.

The quote shows that Per is not afraid of any consequences of being away from work. Creating a father–child relationship is more important than what a co-worker might think. It should be mentioned that it is a matter of course to take parental leave at his workplace, and that his supervisor allows him to prioritize time with family. In this way, his supervisor can be seen as a norm-setter (Moen 2018).

It is also interesting that he would take parental leave 'no matter what people actually say' and if there was resistance from the workplace, he would go against the norm and practice. It should be said that this informant is the youngest in this study, and it is conceivable that a new generation is joining the financial industry when it comes to the use of the father's quota and parental leave.

### 5.2. From Full Quota to Zero Quota

Thomas is a broker, has three children and chose to use the entire father's quota with his first two children. He describes his leave-time as a nice time, and he recommends that other fathers make use of the father's quota. Still, he chose not to use the rest of his quota when he had his last child. This is because his position in the company has changed and he has become head of the department. If he were gone for three months, the department would lose too many clients according to his understanding:

I found that it was a bad solution for me. A bad solution for me and the office that I would be gone for three months. In relation to the projects we had and . . . as we are not so many, we become vulnerable in relation to it. [..] In addition, I find this NAV system (the Norwegian Labor and Welfare system) difficult.

Based on the foregoing statement, Thomas may seem irreplaceable in his position as a broker due to the size of the department (Nordli 2019). In addition, he has previously stated that only he and one other have long experience in the department, while two others are younger and not as experienced. It may therefore appear that the career logic requirement of experience related to networking and customer relationships makes him indispensable in his position.

*5.3. What Should I Come Back to?*

For some informants, the customer relationship has such a central role for the position that taking the full father's quota is perceived as a risk. Geir chose to withdraw the father's quota with both his children incrementally. He took two weeks of leave, and then he took the rest of it in connection with holidays or planning days in the kindergarten. Like Thomas, he experienced the NAV-system (the Norwegian Labour and Welfare system) as difficult to understand, and therefore he used leave only when he already knew he had to leave work.

> The reason I took leave in connection with holidays is that it is simpler to combine father's leave with vacation time than to take leave from full-time work. For NAV (the Norwegian Labor and Welfare system) is quite like that, you have said that you should have leave and then NAV grants you leave [ ... ] Then it became much better for me to use up the leave when I knew I would be gone anyway. Then I could either use the father's quota [ ... ] or take a vacation.

Geir, a broker, also did not manage to see how he could be away from work for a long time because he thought that the consequences of absence would be too great. His wife also wanted to stay at home with their child as long as possible. Customers can easily be transferred to other employees, so he felt that he had to be available for work so that he would not be replaced. In other words, availability and substitutability are linked in this case. You must be available to be non-replaceable. As one can see, it is to the greatest extent the working conditions and tasks that control his leave of absence. Geir has previously stated the following:

> I often compare it to playing on a football team. As long as you score and the team wins, it is good. But if your team-mates think about bringing in someone who can score even more goals, then they will rather play with him on the team. So it's really a pretty demanding industry [ ... ]. You can see that there is a big turnover in that those who are not good enough are squeezed out. And just as in football, those who are good enough get new opportunities.

At all times Geir felt that he has to show that he is a "player" who can deliver results. If one does not perform well enough, that person can be replaced. This can be seen in the context of why Geir does not want to be away from work over a longer period. This stands in a contrast to the male manager, Harald, who was given a managerial position with greater responsibility when he took the father's quota.

Some of the male brokers in this study felt that they have to prove that they are irreplaceable at their workplace, and for Geir it is really important to be available for work. To be available and to maintain close contact with the customers is the only way to score "goals":

> I think if I had to be away for ten weeks [ ... ] I am not sure what I would come back to. Call it the order book; it would have been empty when I returned. [ ... ] It would take a long time to get back up to marching speed. I do not think that I would be ruined for just ten weeks; I would be ruined for a lot longer after that too. It is almost irrelevant whether the employer makes arrangements or not.

Having a job that involves starting and finishing projects was in this case understood as a barrier to using the father's quota. For Geir, absence for long periods would have had major consequences. Geir is not alone in this understanding of the likely consequences of absence from work. Knut is another father who has chosen to be away from work for a shorter period. At the time of the interview, Knut had not taken leave yet, but he had planned to spend a month at home. In addition, he wanted to have a day off each week. For him it was not possible to take more continuous leave for longer than a month:

... And the reason I don't have more is because [ ... ] I have a job with a close customer contact. I talk with the customer several times a day, and then three months will be too long. A month is the limit.

In this case, it is the customer-contact that determines how much leave he is prepared to take. Like Geir, irreplaceability plays an important role. Placing himself in a position where he is inaccessible to his customers for an extended period of time would prevent the broker from appearing as irreplaceable in his work. Hence, he would lose status and advancement.

## 6. Conclusions

The father's quota which reserves part of the parental leave for the fathers has played an important role in getting more fathers to use parental leave. Earlier research has shown that there has been a lower use of leave among fathers in career jobs. The lower use in this group has been explained by the fact that these fathers are in jobs where career logic indicates the need to be always available and to be seen as irreplaceable. We have chosen to use the two-concept pairs 'replaceable and irreplaceable' and 'available and unavailable' to analyze the fathers' use of the father's quota in two different work organizations in order to see how different institutional logics impact the use of the father's quota.

The findings show that the terms 'replaceable and irreplaceable' and 'available and unavailable' work somewhat differently among male managers in the engineering industry and male brokers in the financial industry. The managers in the engineering industry experience that it is possible to be unavailable and replaceable at work, thus they also choose to make themselves unavailable for work when they use the father's quota. It is important for the male managers to set a boundary concerning work when they are on leave. By making themselves unavailable, they become replaceable at work because the organization has to find solutions without them.

The brokers in the finance industry feel that they are irreplaceable at work and they want to be available for work at all times. The brokers have close contact with clients and they think being away on leave for three months is too long a period. Halrynjo et al. (2019) found that parental leave and absence from work are especially pinpointed as risks in Front End Finance. This is because one has a lot to lose in terms of customers, salary development, bonuses, and career opportunities. In their study, the focus is on mothers and that mothers are at a greater risk of losing their customers and "moving back to start." The findings show that this is also the case among fathers in our study. In theory, it is not problematic among the fathers in this study to give up clients to colleagues, but then one has to start all over again when one is back from leave. Unlike the male managers, they experience that there is a risk to using the father's quota when it comes to work and career. Thus, they make themselves available for work and they continue to be seen as irreplaceable.

This shows that the institutional logics are closely linked to the concepts 'available' and 'unavailable'. To be unavailable during leave time seems to be the logic for the male managers in the engineering industry. This is the type of culture and practice that is encouraged in the financial industry. At the same time, the managers are concerned that the father's quota is available for all the male employees. This sets the framework for which practice and choices other employees should use when it comes to parental leave. To be available for work is important for the male brokers in the finance industry. In spite of a positive attitude regarding the father's quota on the party of the brokers, they still hesitate to use it. By not taking the father's quota or just parts of it, they make themselves more available for work, and this becomes the general practice and the institutional logic within the organization.

One of the brokers stands out and experiences leave as particularly important for creating a good father and child relationship. He does not care what co-workers might think about him taking long leave. This informant is the youngest in this study, and it is possible that a new generation is joining the finance industry and will contribute to a gradual change in the institutional logic when it comes to the use of the father's quota. This is similar to the use made by one of the managers of the father's quota as a protest against the workplace and the prevailing 'old-boys culture'.

We found that the working logic is different in the two groups and organizations studied. The leaders in the engineering industry work as a team—as opposed to the brokers who work more individually. In other words, organizational institutional logics provide guidance on what practices one can have within the given organizations. That is, even if one wishes to use the fathers' quota, it is not necessarily the case that the work is experienced as compatible with practice.

It is important to remember that the fathers in this study belong to the group of fathers who have a low take-up of father's quota. In the case of career fathers with an elite education who are a part of a work force, where the job is seen as an investment in career development, a father's use of leave may have indirect and long-term consequence that can reduce his career opportunities (Halrynjo and Lyng 2013). However, the findings show that there are some variations between the two organizations studied. The institutional logic among the male leaders in the engineering industry makes it possible for the fathers to use the father's quota and parental leave. In contrast to this, the institutional logic among brokers in the finance industry makes it difficult to use father's quota or parental leave. In other words, we can find a change in the use of parental leave in one of the organizations, but not in the other. It may seem as if the institutional logic stands stronger among the brokers in the finance industry than among the workers in the engineering plant. This is a study with few informants, and future research is needed to decide whether we can observe a further normalization process among this group of fathers concerning take up of the father's quota.

**Author Contributions:** Conceptualization, L.V.-M. and E.K.; Data curation, K.N.; Formal analysis, L.V.-M. and E.K.; Investigation, L.V.-M.; Methodology, L.V.-M.; Writing—review & editing, E.K.

**Funding:** This research was funded by the Faculty of Social Science and Technology, Norwegian University of Science and Technology.

**Conflicts of Interest:** The authors declare no conflict of interest.

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
