# Peer review of "Father’s Use of Parental Leave in Organizations with Different Institutional Logics"

_socsci, doi:10.3390/socsci8100294_

Round 1

Reviewer 1 Report

The paper provides interesting results about the differences between industries in the institutional logics related to fathers' use of parental leave in Norway. Two datasets are used: interviews with male managers in the engineering industry and interviews with male brokers in the finance industry.

It seems that these two datasets have been collected separately, using different interview guides but this should be explained more clearly. For the interviews with the engineers, it is mentioned that the interview were semi-structured, and the main topics are mentioned. For the brokers, it is mentioned that the interviews were "qualitative" but no topics are mentioned. Also the analysis methods (such as content analysis, narrative analysis) are not mentioned and should be added.

The results seem to be reported separately for the engineers and the brokers, but this is not explicitly mentioned. The reader keeps wondering about which of the interview excerps are from which industry. This could be solved by modifying the subtitles so that the industry is mentioned, or mentioning the occupation in relation to the names of fathers (this is done for only one of the fathers, Markus).

Some concepts used in the text could be clarified: what is the definition for "career fathers" and "elite education" (in the Norwegian context)?

Author Response

We have now written that the interviews with the engineers and brokers are semi-structured and the topics are mentioned.

We have also included that the article is based on a narrative analysis.

Industry is mentioned in the subtitles when the results are reported.

The definition of career fathers and elite educated fathers is discussed page 3 when we discuss previous research in this field.

Reviewer 2 Report

The article addresses an interesting and socially relevant topic, in as far as it analyses the barriers or acceptance of fathers' quota in the Norwegian parental leave system, a forerunner in the extension of leave rights to fathers.

The paper is well structured and present relevant results. Author should, however, discuss also differences in leave taking capacity in the financial sector depending  on if brokers are placed in 'front office'  or 'middle office' , as having to deal with clients seems to be the key barrier in this case. Further, a closer discussion with results included in Halrynjo et al (2019) study of the financial sector would be enriching. Another aspect that would deserve some more attention is the size of the company, as in larger companies there may be more options to adapt to longer leaves. In the data section, information about the size of the companies where respondents work should be included.

The paragraph situated between lines 451 and 456 is too speculative, as arguing based only in two cases is too risky.

Sentence in lines 443 and 444 doesn´t seem to make much sense.

Author Response

The point about the front and middle office is taken care of on page 7, where we point out that we have chosen to concentrate on the front office group because they are the group that resembles the manager group.

A closer discussion of Halrynjo's results is included on page 16.

The size of the companies is mentioned in the methods part for both groups.

The paragraphs between lines 451 and 456 have been changed in order to emphasize that this is the informant's understanding and not necessarily the fact.

Sentence in line 443-444 are changed in order to make more sense.